# Dual adversarial model for generating 3D point cloud

## Abstract

Effectively and accurately synthesizing point clouds is yet a challenge task due to the high-dimensionality of the observation space. In this article, we focus on the generative model to capture the informative intrinsic structure in the latent low-dimensional space and the diversity in the ambient space. We propose a new dual adversarial model which combines Generative Adversarial Network (GAN) and AutoEncoder (AE) with well-designed loss. Extensive experiments demonstrate the effectiveness of the framework on 3D point clouds synthesis. The dual adversarial model achieves the state-of-the-art performance comparing with several well-known point clouds generation models. We also perform experiments on MNIST dataset that exhibit the competitive performance on 2D data.

## 1 Introduction

Point clouds are widely used in many fields, such as vision, 3D object detection and so on. In particular, 3D point clouds can represent geometric details of the object, and suitable for simple geometric transformation. However, there exist significant disadvantages in the point cloud data. For instance, data points associated with objects in the distance are difficult to identify due to sparsity. Moreover, it is not easy to design suitable generative models because of the high-dimensionality of the observation space Gumhold et al. (2001); Daniels et al. (2007); Achlioptas et al. (2018). Effectively and accurately synthesizing point cloud data is still a challenge task.

A qualified point cloud data generation network should satisfy the following conditions: 1) It should be able to synthesize 3D objects that are both highly varied and vivid, which means that there need to be fine details in the generated examples Wu et al. (2016). 2) The distribution of the data generated should be similar to that of the original data. If the dimension of the generated sample is higher than that of the raw data, the additional generated portion must follow the potential distribution of raw data points. In the past few years, many researchers have developed a lot of meaningful work in this direction, such as 3D modeling and synthesis Carlson (1982); Van Kaick et al. (2011). However, many of these traditional methods synthesize new objects by borrowing components from existing CAD model libraries. Therefore, synthetic objects may look realistic, but they are not novel in concept in fact.

In recent years, the development of deep learning has provided a new way to solve this problem. Deep learning can well learn the characteristics of data in a domain by self-updating. Girdhar et al. (2016); Qi et al. (2016) proposes voxelized-object based method in learning deep object representations, while Kalogerakis et al. (2017); Su et al. (2015)(add) from the perspective of view-based projections. Besides these, graph methods Henaff et al. (2015); Defferrard et al. (2016); Yi et al. (2017); Bruna et al. (2013) are also considered for processing algorithms for 3D data. The method of generating point clouds at this stage no longer retrieves them from the object database, instead, most of them synthesize new objects based on learning objects. It is a more difficult task compared to operating on 2D data because of the higher data dimension and fewer data features.

Generative Adversarial Network (GAN) Goodfellow et al. (2014) is the most widely used model in the current generation domain and has achieved great success in 2D image generation field. Wu et al. (2016) first applied this network to the 3D point cloud generation field, and named it as 3D-GAN and achieved great results. It introduces an adversarial discriminator to differentiate whether an object is synthesized or real, which may own the potential to capture the structural difference of two 3D objects.

Although 3D-GAN has been proved to be successful, it still has some problems. Because of the fact that GAN training is difficult and unstable, its huge network parameters drag down the calculation speed Salimans et al. (2016).Groueix et al. (2018); Li et al. (2019) by adding prior knowledge helping generating target data, Zhao et al. (2018) applied this learned prior to the generation of discrete data, and then Achlioptas et al. (2018) adopts the idea of adversarial in latent space, but the specific approach is completely different. The former proposes a workflow that first learns a representation by training an AE with a compact bottleneck layer, then trains a plain GAN in that fixed latent representation. The two parts are trained separately, and their sampling and coding representations are different thus they can't be translated or interpreted within both spaces; the latter has used the Adversarial Autoencoder (AAE) Makhzani et al. (2015) for 3D generation, having the distribution of latent space of autoencoder trained on point cloud data close to the prior distribution, such as standard gaussian distribution.

The manifold hypothesis indicates that the real-world data in a high-dimensional observation space always tends to live in the vicinity of an intrinsic low dimensional manifoldRifai et al. (2011). In this paper we propose a *dual generation model*, which can fully make use of the characteristics of data in low-dimensional latent space to help the network better learn and generate point cloud data. Our contributions are mainly from the following aspects:

- Compared to the literature study on the latent space of raw data, our method, through reasonable theory explaining and experimental verification, proposes dual generate adversarial networks architecture, from which the two networks are divided into the primary and secondary part. The secondary network assists the primary network by learning that the data is compressed to features on different dimensions, and our results exceed the same type of articles that focus on hidden space learning Achlioptas et al. (2018).

- Our model consists of three parts and can be trained end-to-end, what's more, each part interacts and restricts each other, which can be trained efficiently and steadily without using the pre-trained weights related to point cloud data and only adopt the normal random initialization method. The loss function is designed in the light of the function of each part in the network, and we managed to find the best strategy of each part's iteration renewal through experiments.

- The framework we proposed on learning different low-dimension channel can be transferred to other vision tasks easily. We do experiment on the MNIST dataset to demonstrate its implementability on 2D image tasks.

## 2 RELATED WORK

In this section, we mainly introduce the necessary research backgrounds of this paper, including the characteristics of point cloud data and some related work.

**Point cloud** The point cloud is a collection of massive points that form characteristic of the target's surface represents the shape of the object in 3D space. It consists of many $x$, $y$, $z$ coordinates points, which have a matrix structure of $N \times 3$, where N is the number of points in a set. Compared with image data, there are few information and extractable features as to point cloud data, so how to effectively learn and represent the features of point cloud data has become the focus of current research. Wu et al. (2015) took an input point cloud as the voxelized representation, other methods like Maturana & Scherer (2015) using 3D-CNN to calculate occupancy grid. Qi et al. (2017) proposes the PointNet architecture, it can solve the difficulty of dealing with unordered sets by introducing permutation-invariant function instead of convolutions.

**Deep learning for generating 3D data** Learning and generating 3D object is an essential task in the graphics and vision community. In 3D generating field, Wu et al. (2016) proposes the 3D-GAN, which can generate 3D object from a probabilistic space by leveraging recent advances in volumetric convolutional networks and generative adversarial nets. Achlioptas et al. (2018) designed a generative model called latent-GAN which is composed of two procedures: first using the point cloud data as input to train an Autoencoder (AE)Rumelhart et al. (1988), and then using the pre-trained encoder output as the label to train GAN, instead of training on the raw point cloud. Compared with 3D-GAN, its generative ability has improved, but the drawback of this model is that it is not an

end-to-end one. Sampling and coding representations are different and can not be translated or interpreted within both spaces. Our model solves this shortcoming by proposing a dual GAN architecture and the result shows that it can largely improves the generation ability.

## 3 BACKGROUND

In this part we give the necessary introduction on some basic concepts which we will use in the rest of this paper, including point cloud metrics and some fundamental building blocks.

**Autoencoder**   Autoencoder (AE) is a multi-layer neural network that can learn the efficient representation of input data through unsupervised learning. It is composed of two parts: Encoder and Decoder. The networks' width of the former part gradually becomes shallower as the network becomes deeper, while the latter part is reversed, as is shown on the right. The Encoder learns a low-dimensional latent representation $z$ of the input data $x$ and the Decoder restores $z$ to original $x$.

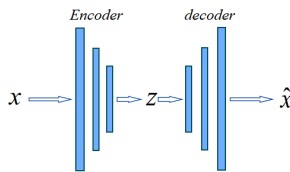

**Generative Adversarial Network**   Generative Adversarial Network (GAN) is the most widely used generative model today. It consists of a model *generator* (G) and a model *discriminator* (D) and establishes a min-max adversarial game between these two models. The generator tries to get the function $G(z)$ that maps $z$ sample from the prior $p(z)$ to the data space. The discriminator tries to distinguish the real data between the generated output. The loss function of this game can be expressed as:

$$\min_G \max_D E_{x \sim p_{data}}[\log D(x)] + E_{z \sim p(z)}[\log 1 - D(G(z))] \tag{1}$$

The model mainly includes two parts of training process:1) Train the discriminator to distinguish the true label between the fake data generated from sample noise by generator. 2) Train the generator to generate samples which aims to fool the discriminator.

**Point Cloud Metrics**   We adopt two permutation-invariant metrics to compare unordered point sets, which was proposed by Fan et al. (2017).
The *Earth Mover's distance* (EMD): Introduced in Rubner et al. (2000), it is a metrics of a transpotation problem based on the minimal cost to transform one distribution into the other. For two equally sized subsets $s_1 \subseteq R^3$, $s_2 \subseteq R^3$, their EMD is defined by:

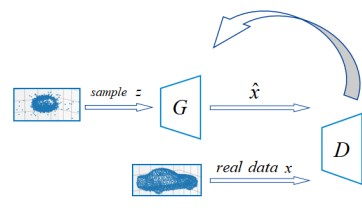

$$d_{EMD}(S_1, S_2) = \min_{\emptyset:S_1 \to S_2} \sum_{x \in S_1} \|x - \emptyset(x)\|_2 \tag{2}$$

where $\emptyset(x)$ is a bijection, EMD is differentiable almost everywhere as a loss.

*Chamfer pseudo-distance* (CD): It measures the squared distance between each point in one set to its nearest neighbor in another set, and the function is shown below:

$$CD(S_1, S_2) = \sum_{x \in S_1} \min_{y \in S_2} \|x - y\|_2^2 + \sum_{y \in S_2} \min_{x \in S_1} \|x - y\|_2^2 \tag{3}$$

Compared to EMD , CD is more efficient to compute as a loss and is totally differentiable.

## 4 DUAL GENERATION MODEL FOR LEARNING REPRESENTATION

In this section we first describe the architecture of our model and the motivation why we design such a network. Then we give a brief introduction to some similar previous work on generating point cloud, and make a comparison of these models to our model. Finally we present a detail explanation of the loss function in our model and how to train it.

### 4.1 OUR ARCHITECTURE

The dataset we use to present our result is ShapeNet Chang et al. (2015), the dimension of each sample is $2048 \times 3$. Inspired by the article Achlioptas et al. (2018) which proposes that using the latent codes instead of raw point data to train 3D-GAN can achieve more precise metrics and less network computing consumption, it is sensible to make a hypothesis that if learning the latent codes can achieve better result, the generator, in the process of generating images from noise, must have a certain dimension that the feature map generated in this dimension is the effective encoding of the raw data. In other words, the function of the latter multiple layers is simply to decode this feature.

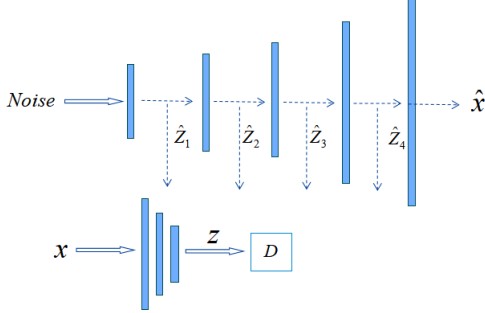

Figure 1: The architecture of our assumption

As the figure shows above, the top of figure is the generator architecture, dimensional transformation occurs at the dotted line. $x$ represents the raw data while $D$ stands for discriminator, $\hat{z}_i$ ($i$=1, 2, 3 ,...) stands for the hidden output derived from each dimension of the generator. According to the above hypothesis, we may make an assumption that $\hat{z}_i$ ($i$=1 or 2 or 3,...) is an effective compress of the raw data, which will be evaluated at the next section, then we can achieve high level on each indicator by adding an adversarial branch to the 3D-GAN in appropriate dimension and precisely designing loss function and updating strategy of each module. From what has been discussed above,

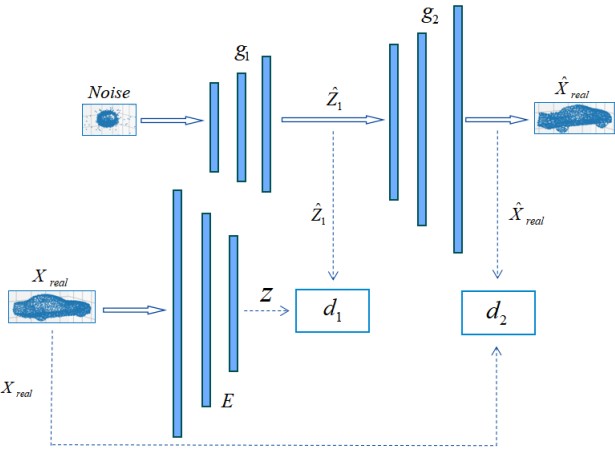

Figure 2: The architecture of our approach

we propose a dual generative model which is presented in Figure 2. To make it easier to illustrate, the modules in the structure are named separately as $g_1$, $g_2$, $d_1$, $d_2$ and $E$. $g_1$ and $d_1$ are combined as generative adversarial model $G_1$ and they are against each other in latent space. The raw data are trained as the label of $G_1$ after being encoded by module $E$ while $g_1$, $g_2$, $d_2$ are combined as the 3D-GAN model named $G_2$ as figure shows above. To expressly state it, we name the lower part of the structure as $E$ which is similar to the model encoder in order to be easily understood. In fact, there is no real decoder in this model to form the autoencoder. During the training process, $G_1$ will

provide auxiliary training for $G_2$, we can consider $G_1$ as a regularization for $G_2$, since it helps $G_2$ more fully trained and better capture the distribution of the raw data.

Our structure combines the advantages of many classic structures in forward propagation, if we focus on module $E$ and $g_2$, this branch is like autoencoder; if we pay attention to $g_1$, $g_2$ and $d_2$, we will get the structure of raw GAN; if we only use $g_1$, $d_1$ and $E$, they make up latent-gan proposed by Achlioptas et al. (2018). In other words, our model can be transformed into the three frameworks mentioned above by some tricks, so we can achieve better performance on their basis.

### 4.2 LOSS DESIGN AND TRAINING PROCEDURE

In this section, we will show our design of the error function, as well as the update strategy of each part. We use the loss function $\mathcal{L}_{GAN}$ and $\mathcal{L}_{AE}$ referred to in Section 3 to compose our loss. The following statements are details of the loss function in each module. We use leaky-ReLU Maas et al. (2013) as our active function and there is no batch-norm.

- $E$: Since the module E is to make an effective dimension reduction for the raw point cloud data, we use the $d_{EMD}(\hat{X}_{real}, X_{real})$ or $CD(\hat{X}_{real}, X_{real})$ to update its parameters. $\hat{X}_{real}$ is the output of $G$, so the update process of this module is in cooperation with the auxiliary generator training. The loss function of module E is named as $\mathcal{L}_E$.

- $G_1$: The purpose of module $G_1$ is to constrain the output of the hidden layer in the entire model, so that the output distribution of the hidden layer is close to the output distribution after the dimension reduction which is operated by module $E$. We use loss function $\mathcal{L}_{GAN}(noise{\rightarrow}z)$ to train this module.

- $G_2$: $G_2$ is a typical 3D-GAN structure, and its loss function must contain $\mathcal{L}_{GAN}(noise{\rightarrow}\hat{X}_{real})$. Inspired by Makhzani et al. (2015) & Larsen et al. (2016), the difference between the generated data and the raw data can be considered as a constraint while updating generation network. Therefore, the loss function of this module is $\mathcal{L}_G = \mathcal{L}_{GAN}(noise{\rightarrow}\hat{X}_{real}) + \lambda\mathcal{L}_E(0 <\lambda< 1)$.

The flowchart of the updating process is as follows:

---
**Algorithm 1** Training the Dual Generation Model

---
initialize parameters of each module $\theta_{G1}, \theta_G, \theta_E$
**repeat**
    $X_{real} \leftarrow$ random mini-batch from dataset
    $noise \leftarrow$ sampled from prior distribution $\mathcal{N}(0, I)$
    $z \leftarrow E(X_{real})$
    $\hat{z} \leftarrow G_1(noise)$
    $\hat{X}_{real} \leftarrow G_2(noise)$
    $\mathcal{L}_E \leftarrow d_{EMD}(\hat{X}_{real}, X_{real})$ or $CD(\hat{X}_{real}, X_{real})$
    update parameter $\theta_E \leftarrow \nabla_{\theta_E}\mathcal{L}_E$
    $\mathcal{L}_{G_1} \leftarrow \mathcal{L}_{GAN}(noise{\rightarrow}z)$
    update parameter $\theta_{G_1} \leftarrow \nabla_{\theta_{G_1}}\mathcal{L}_{GAN}(noise{\rightarrow}z)$
    update parameter $\theta_{G_2} \leftarrow \nabla_{\theta_{G_2}}(\mathcal{L}_{GAN} + \lambda\mathcal{L}_E)$
**until** algorithm convergence

---

## 5 EVALUATION

In this section we will present our results and the enhancement on 2D and 3D data. In 3D case, if we want to evaluate how well the fake data generated by our model matches the given set, we need a comparison to evaluate the faithfulness and diversity of a generative model, referring to Achlioptas et al. (2018), we will use the following metrics:

*Coverage* It is defined as the fraction of the point clouds in $B$, from which we first find its closest neighbor in $A$. It has two forms: COV-CD and COV-EMD, depending on which metrics in Section 3

we use to compute point-set distance.The value of coverage will highly increase if most of the point clouds in $B$ are more matched to point clouds in $A$.

*Minimum Matching Distance(MMD)*  Unlike COV, which only focus on the closest point cloud and isn't the representative of fidelity, we consider MMD as a metric to report the average of distance in the matching. Just like COV, we can use either of the structural distance, so there are MMD-CD and MMD-EMD. The lower the value of MMD is, the stronger the model generation ability will be.

*Jensen − Shannon Divergence(JSD)*  The Jensen-Shannon Divergence (JSD) is used to measure the difference between two empirical distributions. Assume that we have two distribution $P$ and $Q$, so JSD is defined as:

$$JSD(P\|Q) = \frac{1}{2}KL\left(P(x)\left\|\frac{P(x)+Q(x)}{2}\right.\right) + \frac{1}{2}KL\left(Q(x)\left\|\frac{P(x)+Q(x)}{2}\right.\right) \quad (4)$$

where $KL(P(x)\|Q(x))$ means the KL-Divergence between $P$ and $Q$ Kullback & Leibler (1951)

## 5.1 OTHER WORK FOR CONTRAST

Referring to what was described in Achlioptas et al. (2018), we use the following algorithm framework as comparison:

- **Raw-GAN**: It use the basic architecture of GAN which directly generates the raw point cloud data using noise sampled from prior distribution $\mathcal{N}(0, I)$.

- **l-GAN**: Instead of operating on the raw point cloud input, it passes the data through a pretrained autoencoder, and each object class is trained separately with EMD or CD loss. The dimensional of bottleneck variable of AE is 128.

## 5.2 EVALUATING THE LATENT SPACE REPRESENTATION

We use the linear model on latent space which is generated by our model $E$, and its performance can reflect the quality of the latent representation. We train our module $E$ across all shape categories. To obtain features for an input of 3D shape, we feed the point-cloud into module E and then extract the 128-dimensional bottleneck layer vector. The features are processed by a linear classification SVM trained on the 3D classification benchmark of ModelNet Wu et al. (2015). Table 1 shows the comparative results.

| Dataset | SPH | LFD | 3D-GAN | l-GAN | 3D-AAE | ours-CD |
|---------|-----|-----|--------|-------|--------|---------|
| MN10 | 0.798 | 0.799 | 0.910 | **0.954** | - | 0.932 |
| MN40 | 0.682 | 0.755 | 0.833 | 0.845 | **0.847** | 0.840 |

Table 1: The result of classification on ModelNet40 and ModelNet10, from left to right we refer to Kazhdan et al. (2003); Chen et al. (2003); Wu et al. (2016); Achlioptas et al. (2018)

We can clearly see from the table that our method's performance is between the performance of l-GAN and 3D-GAN. This result is very consistent with our previous inferences. The results of l-GAN's latent space will be directly used for the generation of point cloud images, but the representation of our latent space is only used as an assistance for the main framework. Our results, better than 3D-GAN, also prove that the generation framework, after adding the adversarial branch, has better representation in the latent space.

## 5.3 EVALUATING THE RECONSTRUCTION

In this section, we evaluate the reconstruction capabilities of the proposed autoencoder using unseen test examples. We confront the reconstruction results obtained by the E model and use G2 model as decoder with our approaches to examine the influence of a prior regularization on reconstruction quality. In Table 2 we show the MMD-CD and MMD-EMD value between reconstruction and corresponding ground-truth in the dataset of chair object class.

| Method | JSD(tr) | JSD(te) | MMD-CD | MMD-EMD |
|--------|---------|---------|--------|---------|
| AE-CD | **0.0028** | 0.0067 | **0.0004** | **0.0527** |
| ours-E | 0.0026 | **0.0067** | 0.0011 | 0.0562 |

Table 2: The result of latent space reconstruction ability (tr:train-split, te:test-split)

The result in Table 2 shows that our end-to-end model can approximate result from l-GAN when trained by step. This encouraging result indicates that the generation ability of this network will probably exceed the result of l-GAN step-by-step generation. We will introduce the experiment of this part in the next section.

## 5.4 EVALUATING THE GENERATION

We train and compare a total of five generative models on the point clouds data of car category. The model $E$ is established with 128-dimensional bottleneck, and trained with CD or EMD loss respectively. The training procedure will be stopped once our model's performance on the validation dataset is poor. According to the loss used by module $E$, we have two models of evaluating, which are DG-CD and DG-EMD. We also train 3D-GAN directly on the raw point cloud data as comparison.

| Method | JSD | MMD-CD | MMD-EMD | COV-EMD | COV-CD |
|--------|-----|--------|---------|---------|--------|
| R-GAN | 0.1764 | 0.0020 | 0.1230 | 19.0 | 52.3 |
| l-GAN(CD) | 0.0486 | 0.0020 | 0.0796 | 32.2 | 59.4 |
| l-GAN(EMD) | 0.0227 | 0.0019 | 0.0660 | 66.9 | 67.6 |
| DG-CD | 0.0135 | 0.0017 | 0.0632 | 56.3 | 61.7 |
| DG-EMD | **0.0080** | **0.0013** | **0.0581** | **68.2** | **68.5** |

Table 3: Evaluating 5 generators on test-split of car dataset, for our model we did five experiments and reported its average value

In two different sets of experiments, we measure how well the distribution of the generated samples resembles both the train and test splits of the ground truth distribution, by using our models to generate a set of synthetic point clouds and employing the metrics which is introduced above to compare with the train or test set distributions respectively. All experimental results are shown in Table 3, and it can be seen that our method has achieved the best result on all five indicators, which means that in the algorithm based on generative adversarial network structure, our method has achieved the-state-of-art performance. Figure 3 is a comparison of the results of our method with the results of the R-GAN in generating point cloud data. We can clearly see when generating the same dimensional point cloud data, our method is more concentrated and its contour is more obvious. Figure 4 shows some other examples and results.

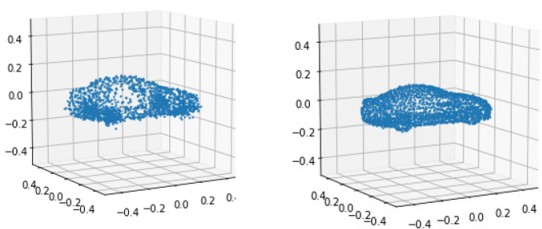

Figure 3: Left: Image generated by R-GAN     Right: Image generated by our method

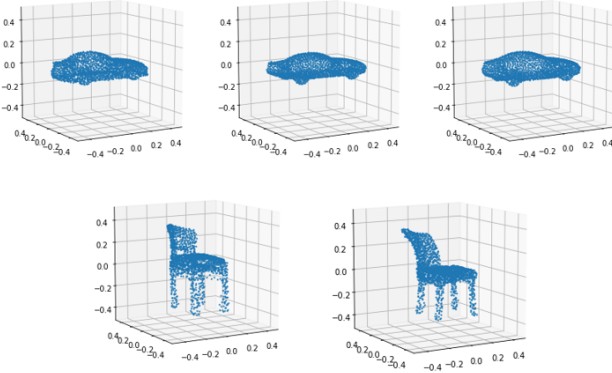

Figure 4: Some results generated by our method

## 5.5 2D RESULT

To prove the mobility of this structure, we did some experiments on the 2D image dataset. We use our method to generate MNIST dataset and compare it with the result of Goodfellow et al. (2014), we can see that the edges of the images we generate are sharper and the texture of the picture is more clear.

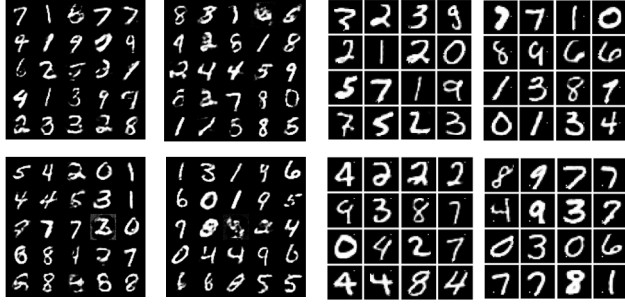

Figure 5: Left: Image generated by GAN       Right: Image generated by our method

| Model | Test Error @50K samples |
|-------|------------------------|
| Real Data | 3.1% |
| GAN | 6.28% |
| Ours | 3.23% |

Table 4: Nearest neighbor classification results

We evaluated the model using a nearest neighbor classifier comparing real data to a set of generated conditional samples, we can see from Table 4 that our approach outperforms the original GAN.

## 6 CONCLUSION

In this paper we propose a dual generation architecture for learning and generating 3D point cloud data, and our experiments show that learning the characteristics of raw input data in different coding-and-low dimension can indeed help the GAN to generate object. Compared with other generative frameworks, our model is an end-to-end architecture which has better performance and higher efficiency. Although our experiment result may not be sufficient due to limited time, the insight of our work is worth for further study and discussion, what's more, the framework based on our learning and representing latent space feature will surely achieve a better result in the future.

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

## A   How $\lambda$ affect the result

In section 4.2, we denoted $\mathcal{L}_G = \mathcal{L}_{GAN}(noise \rightarrow \hat{X}_{real}) + \lambda \mathcal{L}_E (0 < \lambda < 1)$. Different $\lambda$ will lead to different results as figure shows below.

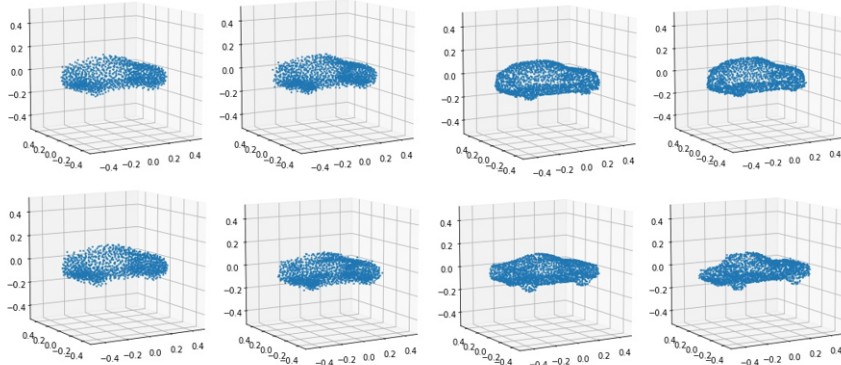

Figure 6: $\lambda$ from left to right is 0.95,0.1,0.01,0.001

From the figure shows above we can see that when $\lambda$ is large, shown in the leftmost column above, the generated image edge is not clear enough and is of less diversity; when $\lambda$ is small, shown in rightmost column, we observe that the generated image is slightly distorted. So we adopt $\lambda = 0.01$ in our experiments.

