# OpenReview forum: "DUAL ADVERSARIAL MODEL FOR GENERATING 3D POINT CLOUD"
_ICLR.cc/2020/Conference — Reject_

### Official Review · AnonReviewer3 · 2019-10-22
**Official Blind Review #3**

**Rating:** 1

**Review:**

The paper propose a dual generator approach for learning to generate point clouds.  The proposed algorithm is a adversarial autoencoder with learned prior generation of point clouds. From this point of view, the novelty of the paper is limited, also, the authors failed to discuss all the related works properly.

1. The problem of the point cloud generation of Achlioptas et al., (2018) is it can't generate arbitrarily many number of points. This issue has been discussed and addressed in those works

* Li et al., Point Cloud GAN, ICLR workshop, 2019.
* Yang et al., FoldingNet: Point Cloud Auto-encoder via Deep Grid Deformation, CVPR, 2018
* Groueix et al., AtlasNet: A Papier-Mâché Approach to Learning 3D Surface Generation, CVPR 2018
* Yang et al., PointFlow: 3D Point Cloud Generation with Continuous Normalizing Flows, CVPR 2019

With stronger prior in the generation process is even studied
* Groueix et al., 3d-coded: 3d correspondences by deep deformation, ECCV, 2018
* Li et al., LBS Autoencoder: Self-supervised Fitting of Articulated Meshes to Point Clouds, CVPR 2019

2. The two generator approach is nothing new.  The first generator g_1 is the same as the "learned prior". There are many related works,  e.g.
* Tomczak et al., VAE with a VampPrior, AISTATS 2018
* Zhao et al., Adversarially Regularized Autoencoders, ICML 2018

3. The paper is not well written, there are several wrong formats of citations and some typos, especially on math symbols, which hinders the reading.  (e.g. G is undefined in Algorithm 1)

**Experience Assessment:**

I have published one or two papers in this area.

**Review Assessment: Checking Correctness Of Derivations And Theory:**

I carefully checked the derivations and theory.

**Review Assessment: Checking Correctness Of Experiments:**

I carefully checked the experiments.

**Review Assessment: Thoroughness In Paper Reading:**

I read the paper thoroughly.

---

> ### Author Response · Authors · 2019-11-15
> **To Reviewer#3**
>
> Q:
> The paper propose a dual generator approach for learning to generate point clouds.  The proposed algorithm is a adversarial autoencoder with learned prior generation of point clouds. From this point of view, the novelty of the paper is limited, also, the authors failed to discuss all the related works properly.
>
> A:
> We must say with regret and respect that Reviewer#3 has unfortunately missed the essence of the article by a large margin. The proposed DUAL ADVERSARIAL MODEL is anything but a more general framework to synthesize data (as Reviewer#1 pointed out). Our approach is absolutely not “a(n) adversarial autoencoder with learned prior generation of point clouds”.
>
> We were highly motivated by the well-known manifold hypothesis. The manifold hypothesis indicates that real-world data in a high-dimensional observation space always tends to live in the vicinity of an intrinsic low dimensional manifold (usually termed low dimensional representation).
>
> Specifically, the set of images associated with the same class always forms a sub-manifold, i.e. a region with dimensionality lower than the original space. Furthermore, for an image dataset with multiple classes, the samples associated with different classes are likely to concentrate along distinct sub-manifolds (also low dimensional), which are separated by low density regions of the observation space. The manifold hypothesis is the underlying assumption of a collection of methodologies, e.g., manifold learning.
>
> From the manifold hypothesis as mentioned above, we know that the sample complexity depends heavily on the “intrinsic” dimensionality, but not the “ambient” dimensionality. Roughly speaking, a normal sample could be divided into two different components - principal component and minor component. The principal component consists of the projection onto the space tangent to the class supporting sub-manifold, which captures most invariance associated with the samples of the class. The component lying in the space perpendicular to the tangent space controls the diversity of the sample and has a relatively minor amount of energy.
>
> Obviously, the roles of the two generator-discriminator pairs (g_1, d_1) and (g_2, d_2) in DUAL ADVERSARIAL MODEL are to capture the informative intrinsic structure (principal component) in the tangent space and the diversity (minor component) in the ambient space, respectively. We conduct extensive experiments to validate the effectiveness of the generator-discriminator pairs (g_1, d_1) in the intrinsic space and (g_2, d_2) in the ambient space. The experimental results also show the competitive performance of the proposed DUAL ADVERSARIAL MODEL for the synthesis of diverse 2D and 3D data with satisfied quality.
>
> As such, we feel strongly that the article is in many aspects up to the standards of ICLR. The comment from Review#3 is a surprise to us. A re-examination of the article is sincerely requested. Thank you for your consideration.

---

> > ### Author Response · Authors · 2019-11-15
> > **To Riewer#3**
> >
> > Q1(1). The problem of the point cloud generation of Achlioptas et al., (2018) is it can't generate arbitrarily many number of points. This issue has been discussed and addressed in those works ...
> >
> > A: No, we don’t think so. We also noticed that the method in Achlioptas et al., (2018) cannot generate arbitrarily many number of points. However, this problem is out the range of our work. In this paper we focus on how to simultaneously capture the informative intrinsic structure in the latent space and generate 3D shape in the ambient space. They are totally two different issues in 3D point synthesis.
> >
> > Q1(2) && Q2:With stronger prior in the generation process is even studied &&The two generator approach is nothing new.  the first generator g_1 is the same as the "learned prior". There are many related works...(these two question express the same query so we will answer them together)
> >
> > A: No, Reviewer#3 has misunderstood the roles of the two generator-discriminator pairs (g_1, d_1) and (g_2, d_2). We don’t employ (g_1, d_1) to learn any prior. It is easily seen that the proposed framework is different greatly than the existing two generator approaches. As mentioned earlier, the proposed framework indeed makes use of (g_1, d_1) to capture the informative intrinsic structure in the low-dimensional tangent space. During training, the two generator-discriminator pairs (g_1, d_1) and (g_2, d_2) are updated alternatively. What’s more, all of the modules in the proposed framework are jointly trained for the end-to-end data synthesis. To the best of our knowledge, the proposed architecture is new and unique.
> >
> > Q(3): The paper is not well written, there are several wrong formats of citations and some typos, especially on math symbols, which hinders the reading.  (e.g. G is undefined in Algorithm 1)
> >
> > A: Thanks for pointing out the typos. We did double check and made the corrections.

---

### Official Review · AnonReviewer2 · 2019-10-26
**Official Blind Review #2**

**Rating:** 6

**Review:**

Summary:

The paper focuses on designing a generative framework for 3-D point data clouds. These point clouds correspond to objects shapes in 3-dimensions. According to the paper, previous approaches for generating such 3-D point clouds involved autoencoder and GANs used separately. The authors propose a framework combining both autoencoder and GAN in a single network. The authors claim that part of the network learns effective latent space embedding for 3-d point clouds corresponding to different objects and thus the entire network is more effective in generating 3-d point cloud. Experimental results are presented to support the claims for efficient embeddings, and better generation of 3-d point clouds.

Decision

The paper has some lacking in explanation. I am particularly interested in the answers to the following questions:
1.	The encoder module (denoted as E in the paper) uses a loss function that apparently does not involve the encoder output (z in the paper). How the module weights can be updated this way is unclear.
2.	G1 is updated twice in each iteration, according to algorithm 1 in paper (once during update of 𝜽G1 and 𝜽G). How the generator G1’s output manages to converge to output z from encoder E is unclear.
3.	The authors claim that the proposed model has both better performance and efficiency. Although experimental results are provided to support claim for better performance, none have been offered for efficiency claim. (calculation of EMD distance seems to be very expensive computationally)

Feedback

For section 5.5, a quantitative analysis might make the claim for portability of the framework stronger. In paragraph 1 of introduction, disadvantage of 3d point cloud data can be better explained.


**Experience Assessment:**

I have read many papers in this area.

**Review Assessment: Checking Correctness Of Derivations And Theory:**

I carefully checked the derivations and theory.

**Review Assessment: Checking Correctness Of Experiments:**

I carefully checked the experiments.

**Review Assessment: Thoroughness In Paper Reading:**

I read the paper at least twice and used my best judgement in assessing the paper.

---

> ### Author Response · Authors · 2019-11-15
> **To Reviewer#2**
>
> Thank you for your review, and we hope that our revisions address your concerns.
> Q1:
> The encoder module (denoted as E in the paper) uses a loss function that apparently does not involve the encoder output (z in the paper). How the module weights can be updated this way is unclear
>
> A:
> Thanks for reading our paper carefully, and it id indeed a good question. We use the loss between $\hat X_{real}$ and $X_{real}$ to update the parameter E. Although z did not pass g2, we can still do this through some tricks in the code. The purpose of this design is to not only help encoder generate latent code, but also help minimize the distance between representation z and \bar{z} from encoder.
>
> Q2: G1 is updated twice in each iteration, according to algorithm 1 in paper (once during update of G1 and G). How the generator G1’s output manages to converge to output z from encoder E is unclear.
>
> A：
> Thanks. The reason why we let G1 be updated twice in one iteration is as follow: the first time we use Lgan(noise \rightarrow z) to only update the G1 parameter, and the purpose of this step is to make the distribution of $\hat z$ close to that of z; and then we use loss LG to update total model G including G1, in order to let G1 learn global information and increase the diversity of generator.
>
> Q3:
> The authors claim that the proposed model has both better performance and efficiency. Although experimental results are provided to support claim for better performance, none have been offered for efficiency claim. (calculation of EMD distance seems to be very expensive computationally)
>
> A:
> Thanks. The reason why we claim that our model has better efficiency is that our approach is one-stage, and the previous methods with similar ideas are two-stage. The previous approach need to train a model first and then use pre-trained model’s result to train generator, while our model can be trained end-to-end, which will greatly shorten the training time.
>
> Q4:
> For section 5.5, a quantitative analysis might make the claim for portability of the framework stronger. In paragraph 1 of introduction, disadvantage of 3d point cloud data can be better explained.
>
> A:
> Thanks for you suggestion. We have added the quantitative analysis of the 2D results in section 5.5 and modified the paragraph 1.

---

### Official Review · AnonReviewer1 · 2019-10-29
**Official Blind Review #1**

**Rating:** 6

**Review:**

This paper proposed a new dual generation model for learning representation by combining GAN and Autoencoder for 3D point cloud data. The idea is to add a regularization term over the GAN loss, and the regularization term measures the distance between the output of the encoder of Autoencoder and that of GAN by two point cloud metrics: Earth Movers's distance (EMD) and Chamfer pseudo-distance (CD). The experiment shows better representation are learnt by this dual generation model. The paper is in a good writing, and easy to read.

``My suggestions:

1) Although this paper is about 3D-point cloud, the idea itself is also feasible for 2D data. In the experiment, a small section is on 2D image data. In order not to confuse readers, I would suggest writing this paper as a general framework on both 2D and 3D data. Is there any particular reason for this work focusing on 3D point cloud data?

2) In 2D data, how to measure the point-cloud distance?still using EMD and CD?

3) In Figure 2, are you trying to minimize the distance between intermedia representation z and \bar{z} from encoder and GAN respectively as well? I do not find corresponding step in Algorithm 1.

4) How is the lambda changing the results? The regularization is the main idea for this paper, and thus it would be interesting to see how the regularization trades off with the loss.

5) how is EMD's performance compared with CD?

**Experience Assessment:**

I have published in this field for several years.

**Review Assessment: Checking Correctness Of Derivations And Theory:**

I assessed the sensibility of the derivations and theory.

**Review Assessment: Checking Correctness Of Experiments:**

I assessed the sensibility of the experiments.

**Review Assessment: Thoroughness In Paper Reading:**

I read the paper at least twice and used my best judgement in assessing the paper.

---

> ### Author Response · Authors · 2019-11-15
> **To Reviewer#1**
>
> Thank you for your review, and we hope that our revisions address your concerns.
> Q1:
> Although this paper is about 3D-point cloud, the idea itself is also feasible for 2D data. In the experiment, a small section is on 2D image data. In order not to confuse readers, I would suggest writing this paper as a general framework on both 2D and 3D data. Is there any particular reason for this work focusing on 3D point cloud data?
> A:
> A good question, In fact, we have been devoted to the problem of generating 3D point cloud. When I finished my manuscript, I happened to find that the model also works in 2D dataset, but at that time, there was less than one week before deadline, so we didn’t have enough time to do more experiments on 2D dataset and rewrite our paper,thus we just show this part’s results as a supplementary experiment of our model. We will seriously consider your suggestion in rhe future work.
>
> Q2：
> In 2D data, how to measure the point-cloud distance? still using EMD and CD?
>
> A：
> Thanks. We have added the quantitative analysis of the 2D results in section 5.5. In evaluating the model of generating 2D dataset, we usually use Inception Score(IS), 1-NN accuracy and so on. You can see a detailed description of them and some other metrics in [1].
>
> Q3:
> In Figure 2, are you trying to minimize the distance between intermedia representation z and \bar{z} from encoder and GAN respectively as well? I do not find corresponding step in Algorithm 1.
>
> A:
> Thanks. The loss function LGAN(noise\rightarrow z) is used to update parameters $\theta_{G1}$ in order to minimize the distance between intermedia representation z and \bar{z}. We describe this process in the third last line of Algorithm 1.
>
> Q3:
> How is the lambda changing the results? The regularization is the main idea for this paper, and thus it would be interesting to see how the regularization trades off with the loss.
>
> A:
> Thanks, a good question, The results are indeed affected by the value of the lambda, As the lambda increases, the generating results will lose their diversity. The results are shown in the appendix.
>
> Q3：
> how is EMD's performance compared with CD?
>
> A：
> Thanks. Do you mean the performance in section 5.2 and 5.3? This two sections evaluate the AE’s capabilities in our model. We find CD get better performance than EMD, so we use CD as loss function in AE’s training. In section 5.4, we show the performance of these two separately on evaluating the generation ability of total model.
>
> [1] Qiantong Xu, et al. An empirical study on evaluation metrics of generative adversarial networks. https://arxiv.org/abs/1806.07755, 2018

---

### Decision · Program_Chairs · 2019-12-19

**Decision:**

Reject

**Comment:**

The paper introduces a new method for 3d point cloud generation based upon auto encoders and GANs.

Two reviewers voted for accept and one reviewer for outright reject. Both authors and reviewers posted thorough responses. Based upon these it is judged best to not accept the paper in the present. The authors should take the feedback into account in a an updated version of the paper.

Rejection is recommended.